# Valuation of Local Demand for Improved Air Quality: The Case of the Mae Moh Coal Mine Site in Thailand

Worawat Srisawasdi [1], Takuji W. Tsusaka [1,*], Ekbordin Winijkul [2] and Nophea Sasaki [1]

[1] Department of Development and Sustainability, Asian Institute of Technology (AIT), Pathumthani 12120, Thailand; worawat.srisawasdi@ait.ac.th (W.S.); nopheas@ait.ac.th (N.S.)

[2] Department of Energy, Environment and Climate Change, Asian Institute of Technology (AIT), Pathumthani 12120, Thailand; ekbordinw@ait.ac.th

* Correspondence: takuji@ait.ac.th

**Abstract:** While the district of Mae Moh, Thailand has been well known for its atmospheric pollution associated with coal power production, economic assessment of demand for improved air quality has not been conducted to date. This study estimated local residents' individual and aggregate willingness to pay (WTP) for mitigation of atmospheric pollution in Mae Moh using the contingent valuation method (CVM), and analyzed the factors associated with the individual WTP using the bivariate tobit and double-hurdle regression techniques. Primary data were collected through face-to-face interviews with a stratified sample of 200 residents. The hypothetical scenarios used in the CVM module were 50% and 80% mitigation of atmospheric concentrations of major pollutants. The weighted average WTP was found to be THB 251.3 and 307.9 per annum (USD 8.4 and 10.3) for the 50% and 80% reduction scenarios, respectively. The aggregate WTP for the entire population of Mae Moh was THB 10,008,733 and 12,264,761 per annum (USD 336,294 and 412,096), respectively. Education, occupation type, income, expenses, satisfaction with ambient quality, and perceived sources of pollution had significant associations with the individual WTP. The paper concludes by discussing policy implications for atmospheric pollution management and avenues for future research.

**Keywords:** air pollution; air quality; particulate matter; $PM_{2.5}$; $PM_{10}$; contingent valuation; willingness to pay; demand curve; double hurdle model; bivariate tobit

## 1. Introduction

A clean atmosphere is an essential natural resource for sustaining life on the surface of Earth. Today, atmospheric pollution is considered one of the world's most serious environmental concerns associated with health risks, especially in developing countries [1]. The World Health Organization estimated that atmospheric pollution was responsible for over six million premature deaths worldwide in 2016 [2]. The atmospheric concentration of major pollutants such as particulate matter ($PM_{2.5}$ and $PM_{10}$), CO, $SO_2$, $NO_2$, and $O_3$ has been rising at an alarming rate in many parts of the world. One of the major causes of the aggravation of this pollution is fossil fuel combustion activities for energy generation purposes, among other causes. It is estimated that around 70% of the total global energy consumption comes from coal resources [3], whereas excessive fuel combustion processes and inappropriately managed construction activities are the major causes of atmospheric pollution [4]. Epidemiological research on the health impacts of atmospheric pollution has revealed a strong correlation between concentrations of pollutants and respiratory disorders and cardiovascular diseases [5–8].

Thailand is one of the countries where atmospheric pollution has been a persistent concern due to its risk posed on human health. As a developing country and one of the largest economies in Southeast Asia, the nation has been experiencing severe atmospheric pollution for decades in parallel with rapid economic development, exposing its citizens

to adverse effects on health [9]. The epidemiological association with respiratory, cardio-vascular, and cerebrovascular diseases has been established in Thailand as well [10–12]. Pinichka (2017) [13] estimated the burden of diseases attributable to ambient pollution based on the comparative risk assessment (CRA) framework and found that the benefits of a 20% reduction in ambient pollutant concentration could prevent up to 25% of avoidable fatalities each year in all-causes of respiratory and cardiovascular categories. Mueller et al., (2020) [14] attempted to quantify and compare health risks of PM arising from biomass and non-biomass burning sources in northern Thailand and found that there was significant intra-annual variation in $PM_{10}$ concentrations, with the highest concentration occurring during March, coinciding with the peak biomass burning activities, same-day exposures of $PM_{10}$, and the incidence of certain respiratory and cardiovascular outpatient visits. Sources of PM in Thailand are both anthropogenic (e.g., road traffic and industrial emissions) and natural (e.g., forest fires) [15]. It is believed that the significant contributor of PM emissions is biomass burning from wildfires, agricultural slash and burn practices, land clearing, and household fuel combustion [16].

Given the magnitude of the issue, a number of researchers have embarked on quantifying public demand for mitigation of atmospheric pollution by using the contingent valuation method (CVM). Wang and Mullahy (2006) [17] estimated the willingness to pay (WTP) for reducing fatal risk by improving atmospheric quality in Chongqing, China. They found that 96% of the respondents were able to express their WTP and the mean WTP for saving one statistical life (The value of statistical life is the marginal rate of substitution between income and mortality risk [17].) was USD 34,458. Bazrbachi et al., (2017) [18] estimated the public WTP for improving the atmospheric quality by examining the prospect for a transport shift from private vehicles to public transport in Klang Valley, Malaysia. The WTP for continuing to use their private vehicles was MYR 4.99 (USD 1.55) per trip. Sereenonchai (2020) [19] explored the WTP for self-protection and haze management in the Chiang Mai province of Thailand and found that the mean WTP for an N95 face mask was the highest at THB 82.74/piece followed by rural plain areas at THB 55.68/piece, and that the proportion of the respondents willing to pay for a mask was highest in the urban area (59%). Williams and Rolfe (2017) [20] assessed WTP in Queensland, Australia for a reduction in national greenhouse gas emissions by 2020 and found that the value of a hypothetical change from the existing policy to a greater emissions reduction policy was more than USD 400 per capita per year. Guo (2006) [21] assessed health risks related to atmospheric pollution in China and estimated the value of a statistical life with respect to asthma to be USD 2300. However, quantitative assessment of public WTP for mitigation of atmospheric pollution in northern Thailand remains scarce to date. As a result, the economic benefits that would arise from mitigating atmospheric pollution in the said area have not been adequately understood. Moreover, individual factors associated with the WTP have been understudied in the context of values of a clean atmosphere.

The objective of this paper is to estimate the economic values of hypothetical reduction in atmospheric pollutants using the CVM and identify the factors associated with the individual WTP using regression techniques. The paper provides a case study in the Mae Moh district, Lampang Province, Thailand. According to Greenpeace (2017) [22], five provinces with the highest annual average concentrations of $PM_{2.5}$ and $PM_{10}$ were Chiang Mai, Lampang, Bangkok, Khon Kaen, and Ratchaburi. The Electricity Generating Authority of Thailand (EGAT) operates a large lignite coal mine and coal-fired power plants in the Mae Moh District, Lampang province for electricity generation. This operation has resulted in emissions of massive quantities of pollutants into the atmosphere, which directly affected the environments and the health of residents in the nearby areas (ADB, 2008). It is estimated that by 2005, more than 30,000 people (over 80% of the population) in Mae Moh had acquired severe respiratory symptoms due to the inhalation of $SO_2$, $PM_{2.5}$, and $PM_{10}$ emitted from the power plant and coal mine [23]. Following this introduction, Section 2 discusses the literature on the valuation of atmospheric pollution mitigation. Section 3 explains the methodology such as the data collection, background of the study

site, and analytical methods. Section 4 reports the results of the WTP assessment and regression analysis of the factors. Section 5 provides discussions of the findings, followed by Section 6 to conclude the paper by offering policy implications.

## 2. Review of Related Literature

### 2.1. Atmospheric Pollution in Thailand

While rapidly transforming itself from an agrarian to an industrial economy, Thailand has faced increasing levels of atmospheric pollution, which were found to be associated with adverse health effects. Studies have investigated the adverse effects of atmospheric pollution in Thailand. A large-scale population-based epidemiology study with over 26,000 subjects in Thailand indicated that residents living near the petrochemical industrial estate had higher risks of adverse pregnancy outcomes and neuropsychological symptoms, as well as undesirable performance on neuropsychological tests [24]. Another study in Bangkok showed that each 10 $\mu g/m^3$ increase in $PM_{10}$ concentration was associated with a 1.25% rise in all-cause mortality, which was higher than in the three other participating cities (0.53% in Hong Kong, 0.26% in Shanghai, and 0.43% in Wuhan [25]. Guo (2014) [9] studied the association between atmospheric pollution and mortality in Thailand and found significant short-term impacts of all atmospheric pollutants on non-accidental mortality. In particular, pollution with $O_3$ was significantly associated with cardiovascular mortality, while $PM_{10}$ was related to mortality due to respiratory diseases.

Figure 1 shows the different sources of electricity consumed in Thailand. The sources are dominated by fossil fuels, namely, natural gas (72.3%), coal (15.8%), and oil (2.7%), accounting for more than 90% of electricity consumed. Of these, coal-fired power generation is considered to emit the largest quantities of pollutants per unit of power into the atmosphere such as $NO_X$, $SO_X$, HCl, HF, As, and Pb [26]. Currently, Thailand has five coal-fired power plants of over 300 MW and nine smaller plants of around 100 MW, most of which are located in the Mae Moh District, Lampang Province, making it the largest electricity production complex in Thailand with 2400 MW capacity. Furthermore, coal power plants are estimated to account for over 70% of energy-related emissions of $SO_2$ and $PM_{10}$ in Thailand. It is estimated that air pollution emissions from Thailand's coal-fired power plants were responsible for 1550 premature deaths per year, and if the new coal-fired power plants under construction were taken into account, the health impacts could increase up to 5300 premature deaths per year [27].

The largest natural gas power plant in Thailand is the Bang Pakong power plant (1862 MW) in Chachoengsao Province located over 900 km from Mae Moh, which is also the second-largest electricity generation site in Thailand [28,29]. The natural gas power plant nearest to Mae Moh is the Nam Phong Power Plant in the Khon Kaen Province, about 600 km away [28]. The $SO_2$ emissions from coal and natural gas are typically 600 and 10 mg $MJ^{-1}$, respectively, whilst the $NO_X$ emissions are around 300 and 100 mg $MJ^{-1}$, respectively [30]. While pollutants can travel with wind over several hundreds of km from a source [27], the atmospheric concentrations of $SO_2$ and $NO_2$ decrease to 10% in 150 km and further to 1% in 600 km according to Curtiss and Rabl (1996) [31]. Given the long distance, pollutants emitted from the natural gas power plants do not practically affect the atmospheric quality in Mae Moh.

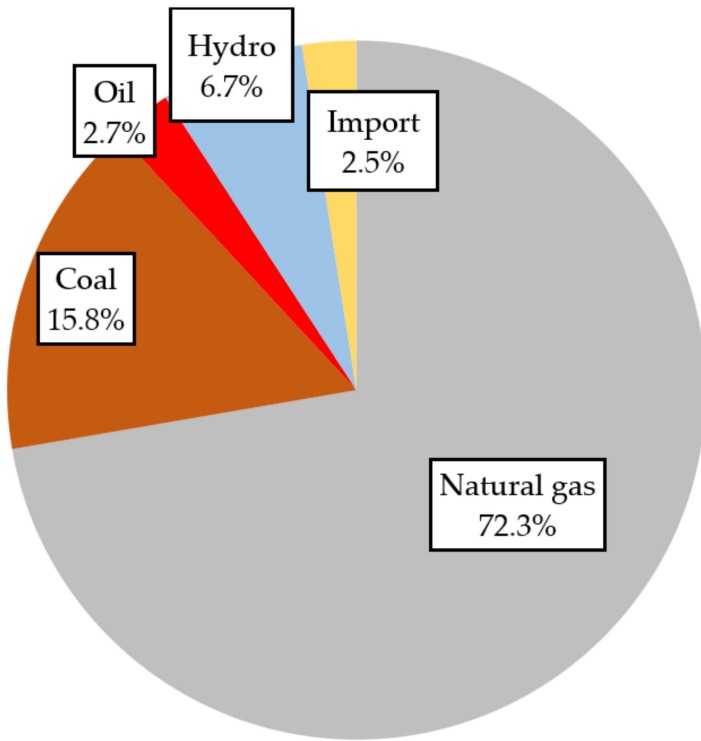

**Figure 1.** Sources of electricity consumed in Thailand. Source: US Energy Information Administration (2019) as cited in [29].

*2.2. Natural Resources Valuation Methods*

Many valuation methods have been adopted in natural resources valuation literature, such as the travel cost method, hedonic pricing method, and appraisal method (Table 1).

**Table 1.** Methods and techniques for natural resource valuation.

| Natural Resource Valuation Technique | Type of Goods Assessed | Type of Value Assessed | Example |
|---|---|---|---|
| Contingent valuation method | Non-market goods or any type of goods | Any type of value | [32,33] |
| Travel cost method | Non-market goods (with related market goods) | Use value | [34] |
| Hedonic pricing method | Non-market goods (with related market goods) | Use value, non-use value | [35,36] |
| Appraisal method | Land/market goods | Use value | [37,38] |
| Market price method | Market goods | Use value | [39,40] |
| Resource replacement cost | Non-market goods (with related market goods) | Use value | [41] |
| Random utility method | Any type of goods | Any type of value | [42,43] |
| Benefit transfer method | Any type of goods | Any type of value | [44,45] |

In general, valuation methods are applied to estimate the demand for the beneficial uses or services enjoyed by individuals and communities. When there are markets where the natural resource goods or services in question are traded, the valuation exercise is relatively straightforward. Examples of natural resources with existing markets are food, fiber, and freshwater. The market-based techniques include the market price approach [40], the appraisal method [38], and the replacement cost method [41]. The disadvantage of market-based techniques is that they require existing markets.

For natural resources without existing markets (non-market goods), other valuation techniques need to be adopted such as the travel cost method [34], hedonic pricing [35,36], and benefit transfer method [44,45]. These methods are indirect market-based methods as

they utilize markets of goods and services that are related to the resource in question. For instance, the travel cost method utilizes the market prices of transport, accommodation, and wage labor to assess the values of a natural resource-oriented tourism destination [34], whilst the hedonic pricing method utilizes the prices of real estate to assess the values of natural resource amenities in the neighborhoods [36]. When non-market goods and services do not have indirectly related markets, the CVM is commonly applied.

The CVM is a survey-based approach to the valuation of non-market goods and services that does not make use of any market [32,33] and it directly queries individuals in order to elicit their WTP for hypothetical improvement in environmental conditions as defined by the researcher. However, the CVM requires certain assumptions, which may not always be realistic. One assumption is that a human is a consciously calculating optimizer who is able to perfectly process any information correctly and react to it immediately [46]. Another crucial assumption is that the environmental damage in question is perceived as having limited boundaries and that monetary payments are able to compensate for the entailed losses. The fact that the CVM is based on respondents' answers to questions (i.e., stated preferences), as opposed to observed actual behavior (i.e., revealed preferences), occasionally creates a controversy [47]. In the economics literature, the estimation of economic values based on how people respond to questions using hypothetical situations has been constantly debated [48].

*2.3. Contingent Valuation Method*

The CVM has been adopted for the economic valuation of a broad range of natural resources especially when there is no market for the resource in question or closely related goods and services [49]. The CVM is regarded as the most flexible valuation method in principle and can be designed to valuate basically any type of environmental goods, services, and externalities [50]. The method also appears to be the only way to elicit economic benefits of the non-use values of environmental goods such as the existence value and bequest value [51].

Open-ended questions have often been used to elicit the WTP, which suffer from the difficulty for respondents in mentioning a specific amount, resulting in missing responses and downward biases in elicited WTP values. An alternative to circumvent this limitation is the bidding approach [52], where individuals are asked whether or not they would be willing to pay an amount designated by the researcher. If the answer is yes (no), the researcher keeps raising (lowering) the value until the respondent's answer changes from yes (no) to no (yes). This approach also has its disadvantage as it is prone to a starting-point bias [52,53]. Furthermore, the repeated questioning may annoy or tire respondents, tempting them to say no (yes) to an ad-hoc amount for the purpose of ending the queries [54]. An alternative to address the limitations of the bidding approach is the payment card method, where researchers present a number of different values on different cards and ask the respondent to pick the amount that best represents his or her WTP [55]. Although this method can overcome the limitations of the bidding approach, it also has drawbacks as respondents might limit their WTP only to one of the values presented on the cards, or the WTP might be outside the range of the printed values [51,55].

Another widely used approach to eliciting the WTP is the dichotomous choice (DC) format. A dichotomous choice payment question asks whether the respondent is willing to pay a certain amount to obtain the good or service [39,56]. As opposed to the bidding approach, the DC format does not use the iterative process but only one or a few values will be designated, which is stochastically analyzed by binary response models [56]. The advantage of the DC format is that it simplifies the cognitive task faced by respondents compared to other elicitation methods that come with more value choices (i.e., open-ended, bidding game, payment card) [57]. It also mimics consumers' behavior in market transactions, where they usually purchase (or not purchase) goods at the specified prices, which raises incentive compatibility [33]. However, the DC format is statistically inefficient, which requires a much larger sample size in order to obtain the same level of precision

as in other elicitation methods, leading to higher costs of data collection [37]. Another disadvantage is that the DC format does not pin down the WTP, but an inference is made that the respondent's WTP is higher or lower than the designated amount, forming intervals around the actual WTP [58].

### 2.4. Valuation of Mitigation of Atmospheric Pollution

To our knowledge, the first CVM study on atmospheric pollution was conducted by Ridker (1967) [59]. While the primary purpose of his research was to valuate damages to soil and household materials, it was his survey questions regarding atmospheric pollution that began to show that the CVM could be utilized as a method for attaching values to atmospheric quality through the psychic costs of pollution. His WTP questions inquired the extent to which people would be willing to pay if they could avoid dirt from the dusty atmosphere. A breakthrough in the use of the CVM occurred through the valuation of the damage caused to ecosystems in Alaska by the Exxon Valdez tanker spill in 1989, for which a group of researchers legitimized the use of the CVM for assessment of environmental damage [60]. Arguably, the CVM is the most popular method for economic valuation of atmospheric quality improvement or pollution mitigation [61–63].

In addition to some of the literature reviewed in Section 1, there are several relevant valuation studies on atmospheric pollution reduction. Wang et al., (2006) [64] presented a scenario of reducing pollution by 50% in Beijing and found that the average WTP was USD 22.94 per year or around 0.7% of the average household income. A study in Jinan, China showed that the average WTP for atmospheric quality improvement was USD 16.05 per year and 59.7% of the respondents expressed positive WTP [65]. Afroz et al., (2005) [66] estimated the WTP for improving the atmospheric quality in Klang Valley, Malaysia, and compared the results from different elicitation techniques, namely, open-ended questions, dichotomous choices, and payment card format. They found that the average WTP values were not significantly different, though the dichotomous choice format tended to yield the highest WTP values. The average WTP was MYR 9.69 (USD 2.75) per 100 L of fuel. The aggregate WTP was estimated to be MYR 0.91 billion (USD 257.79 million) in total per year. Wang and Whittington (2000) [67] found that households in Sofia, Bulgaria, were willing to pay up to 4.2% of their income for the atmospheric quality improvement program. In Cameroon, Donfouet (2015) [68] proposed a hypothetical referendum scenario of reducing atmospheric pollution in Douala by 25% and found that households were willing to pay USD 0.42 per month or 0.2% of household income for the atmospheric quality improvement program. Sun et al., (2016) [69] estimated the WTP for reducing atmospheric pollution in the urban area of China and found that nearly 90% of the respondents were willing to pay for mitigation of pollutants, and the mean WTP was CYN 382.6 (USD 57.6) per annum. Table 2 summarizes seminal examples of the estimated WTP for atmospheric quality improvement or pollution mitigation.

**Table 2.** Empirical studies on economic valuation of atmospheric quality improvement using the contingent valuation method.

| Literature | Scope | Location | Result |
|:---:|:---:|:---:|:---:|
| [64] | Estimated WTP for a scenario of reducing pollution by 50% in Beijing. | Beijing, China | The average WTP was USD 22.94 per year or around 0.7% of the average household income. |
| [65] | The relationship between poor atmospheric quality and residents' WTP for improved atmospheric quality. | Ji'nan, China | 59.7% of the respondents expressed positive WTP, and the average WTP was CNY 100 (USD 16) per person per annum. |
| [67] | The distribution of WTP to pay various prices using a stochastic payment card approach by asking respondents the likelihood that they would agree to pay a series of prices. | Sofia, Bulgaria | Respondents were willing to pay up to 4.2% of their income for atmospheric quality improvement. The income elasticity of WTP was 27%. |

**Table 2.** *Cont.*

| Literature | Scope | Location | Result |
|---|---|---|---|
| [70] | Estimated the WTP amount for reducing the mortality rate for evaluation of a statistical life value | Seoul, South Korea | Monthly average WTP for mortality reduction was USD 20.20 and the implied value of statistical life was USD 485,000. Total damage from $PM_{2.5}$ was USD 1057 million per year for acute exposure, and USD 8972 million per year for chronic exposure. |
| [71] | The WTP for clean atmosphere by applying the CVM for six damage components using the payment card question format. | Various cities of Poland | The annualized median WTP was PLN 96 (USD 25). The mortality component had the highest mean WTP (23.3% of the total WTP). |
| [72] | The adverse health effects of particulate matter pollution. | Pearl River Delta (PRD), China | The total economic loss of the health effects of $PM_{10}$ pollution in PRD was CNY 29.21 billion (USD 4.63 billion) by the CVM method. The economic loss due to premature deaths and respiratory diseases accounted for 95% of the total loss. |
| [69] | Estimated the WTP for reducing atmospheric pollution in the urban areas of China. | Urban areas of China | 90% of the respondents had positive WTP for reducing atmospheric pollution. The mean WTP was CYN 382.6 (USD 57.6) per year. |

In the subject of damage caused by atmospheric pollution, especially morbidity and mortality, the CVM has been employed as the main tool for performing the benefit-cost analysis of pollution control measures, especially in Asia [73]. Lee et al., (2011) [70] estimated the WTP amount for reducing the mortality rate for evaluation of a statistical life value in Seoul and found that the monthly average WTP for 5 out of 1000 mortality reduction over ten years was USD 20.20 and the implied value of statistical life was USD 485,000. The total damage due to the risk from $PM_{2.5}$ inhalation was USD 1057 million per year for acute exposure, and USD 8972 million per year for chronic exposure. Huang et al., (2012) [72] estimated the adverse health effects of PM in the Pearl River Delta (PRD) in southern China and found that the total economic loss of the health effects of $PM_{10}$ pollution in the PRD was CNY 29.21 billion (USD 4.63 billion) and that premature deaths and respiratory diseases accounted for 95% of the total economic loss.

*2.5. Individual Factors Associated with WTP for Atmospheric Pollution Mitigation*

It is important to understand the factors associated with the individual WTP for improvement in atmospheric quality. Past studies revealed that multiple dimensions of factors had statistically significant effects on individual WTP, particularly in China. Wang et al., (2007) [65] found that the WTP for atmospheric quality improvement in Jinan, China, was associated with demographic attributes, whereby the WTP increased with income and level of education while it decreased with household size and age. Liu (2018) [74] assessed the WTP for improving atmospheric quality among manufacturing workers in Nanchang, China, and found that the main factors associated with their individual WTP were residence areas, education level, household income, and travel experiences.

Khuc (2020) [75] used a stratified sampling technique coupled with the CVM with 475 locals in Hanoi, Vietnam, and found that the WTP for atmospheric environmental funds was associated with a set of endogenous and exogenous factors including age group, level of current atmospheric pollution, income, and awareness of environmental protection solutions. Akhtar (2017) [76] assessed the relationship between degraded atmospheric quality and residents' WTP for improved atmospheric quality in the city of Lahore, Pakistan, and found that household income, symptoms of respiratory diseases, and self-observed atmospheric pollution positively impacted the WTP. Gaviria (2013) [77] surveyed individuals

working in downtown areas of Medellin, Columbia, to measure their WTP for a reduction in atmospheric pollution and reported that age, income level, having a symptom/illness, and exposure to pollution (at different levels) had direct effects on the probability of them stating positive WTP.

In Thailand, Naranuphap and Attavanich (2020) [78] conducted a study to investigate the determinants of the willingness to pay for preventing $PM_{2.5}$ problems in Bangkok using the Contingent Valuation Method (CVM). The results showed that the determinants of the WTP were: number of days of exercise more than 5 days/week, number of household members, income, and the impact of air pollution. All four factors were positively correlated to the willingness to pay with a statistical significance level. Two other factors, including number of children in household and number of elderly in household, were negatively correlated to the willingness to pay with a statistical significance level.

## 3. Materials and Methods

### 3.1. Study Site

The study was conducted in the Mae Moh District, Lampang Province, northern Thailand. The district is well known for being the location for the large-scale power plant and the coal mine. The Mae Moh coal mine is considered to be the largest lignite coal mine in Southeast Asia whilst the power plant is the largest coal-fired power plant in Thailand [29]. However, the power plant and the coal mine are considered to be the major source of atmospheric pollution [79]. The pollution caused public health problems as the combustion of lignite coal emits pollutants into the atmosphere such as $SO_2$, CO, and $NO_2$ [80].

Figure 2 shows the map of Lampang Province and the Mae Moh District. The climate of Mae Moh is characterized by the southwest and northeast monsoons. The southwest monsoon produces the hot wet season between March and September, while the northeast monsoon produces the cool dry season between October and February [81]. During the dry season, the Mae Moh basin is influenced by high atmospheric pressure, causing high ambient $SO_2$ concentrations to accumulate quickly. This accumulation exerts severe acute health effects among the local population [81].

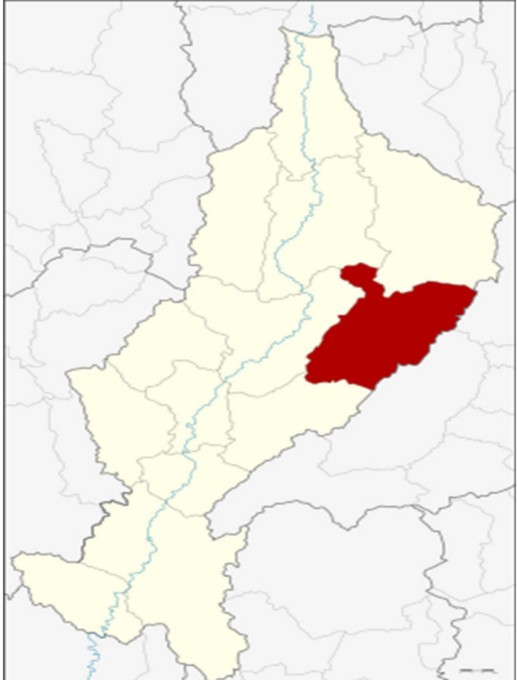

**Figure 2.** Map of Lampang province with the Mae Moh District highlighted in red. Source: [82].

Historically, more than 30,000 local citizens have been affected by the coal-fired power plant in Mae Moh by being displaced and/or acquiring respiratory diseases through inhalation and exposure to high concentrations of $SO_2$ and particulate dust [83]. Furthermore, the fly ash from the power plant affected the crops grown by the villagers. Farmlands were also ruined by acid rain attributed to high concentrations of $SO_2$ in the atmosphere [83]. To mitigate the negative impacts, the EGAT installed pollution control devices at the turn of the century, such as flue gas desulfurization (FGD) and ionizing wet scrubbers [83]. Atmospheric $SO_2$ concentrations in Mae Moh have decreased tremendously since the installation of the control devices [81].

The Pollution Control Department (PCD) reports the Air Quality Index (AQI) for five major pollutants regulated by the National Ambient Air Quality Standards (NAAQS) [27]. At present, the overall AQI for the Mae Moh district is typically between 10 and 20 [83]. In December 2019, the average monthly concentrations for the major atmospheric pollutants were 3 ppb for $SO_2$, 7 ppb for $NO_X$, 1 ppm for CO, 22 ppb for $O_3$, 64 μg/m$^3$ for $PM_{10}$, and 40 μg/m$^3$ for $PM_{2.5}$ [84], indicating that $SO_2$, $NO_X$, CO, and $O_3$ are negligible, while the concentrations of $PM_{2.5}$ and $PM_{10}$ remain worrisome.

### 3.2. Sampling Framework

Survey respondents were selected within the Mae Moh District. The sampling unit was individuals as the valuation was based on individual understanding and perception of the pollution. The sample size was determined by using the Taro Yamane formula [85] as follows:

$$n = \frac{N}{1 + Ne^2} \tag{1}$$

where *n* is the minimum suggested sample size, *N* is the population size in Mae Moh, and *e* is the margin of error. Inserting the total population of 39,831 and applying the 10% margin of error, the minimum suggested sample size was calculated to be 99.7. To absorb the risk of missing observations in certain variables, which would potentially undermine the multiple regression analysis, it was decided to increase the sample size to 200, which corresponded to the 7% margin of error.

The Mae Moh District is divided into five sub-districts (tambon), which are further subdivided into 42 villages (muban). The district has no municipality (thesaban). The quota sampling was used to allocate 40 respondents for each sub-district for face-to-face interviews, followed by systematic random sampling to select and visit the respondents. Table 3 shows the population, sample size, and sampling weight for the five sub-districts of the Mae Moh District. The sampling weight was used later for calculating the weighted mean and aggregate WTP.

**Table 3.** Population and sample size across the five sub-districts of the Mae Moh District. Source: Authors, ref. [86].

| Sub-District | No. of Villages | Population | Sample | Sampling Weight |
|---|---|---|---|---|
| Ban Dong | 8 | 4945 | 40 | 0.124 |
| Na Sak | 9 | 6484 | 40 | 0.163 |
| Chang Nuea | 7 | 5390 | 40 | 0.135 |
| Mae Moh | 11 | 16,034 | 40 | 0.403 |
| Sop Pat | 7 | 6978 | 40 | 0.175 |
| Total | 42 | 39,831 | 200 | 1.000 |

### 3.3. CVM Survey

As atmospheric quality is not traded in markets, the CVM was adopted in this study. The bidding game format was used for elicitation of the WTP as it increases the precision for estimation of the WTP in comparison to the DC approach, incurs less bias than the open-ended question format, and is more operationally practical than the payment card method.

Face-to-face interviews were conducted from 16th to 31st December 2019 at the homes of the respondents as the home was regarded as the place where respondents would feel comfortable and provide well-thought-out responses. The survey questionnaire consisted of three main parts. The first section collected basic socio-economic information, including gender, age, education, area of residence, occupation, and annual income. The second part inquired respondents' perception of the current situation of atmospheric pollution in the areas of residence and work, the relationship between pollution and respiratory diseases, and expenditure on treatment of these diseases. The final section was the contingent valuation instrument based on the bidding game technique. The respondents were given a choice to answer yes or no for the given bid amount. The bid amount started from THB 10 per month with the assumed answer of yes. The amount was increased by THB 10 until the respondents' answers switched from yes to no. The maximum amount that received the answer yes was recorded as the respondent's WTP. If the respondent was not willing to pay any amount, the value of zero was recorded as the WTP.

For the analysis of individual and aggregate WTP, the CVM survey was employed to estimate the values of WTP for 50% and 80% atmospheric pollution mitigation scenarios. Under these hypothetical scenarios, the respondents were provided with adequate information about the question being inquired in order to give a relevant response. However, they should not be overwhelmed with unnecessary detail or subjective information that might bias their valuation [87]. The WTP amount was elicited in a bidding game format. The respondents were given the liberty to bid yes to the amount being presented incrementally until they were no longer willing to pay for the amount. The range and interval of amounts were pre-determined based on the pre-test conducted in Mae Moh using an open-ended question. Accordingly, the WTP amounts were set to range from THB 0 to 200 per month with an equal interval of THB 10.

In light of the unfamiliarity of the respondents with this kind of survey and the need to ensure relevant responses, some essential information was provided and the key premises were explained as follows:

Contextual information:

- The classification of the different major atmospheric pollutants, namely, $PM_{2.5}$, $PM_{10}$, CO, $SO_2$, $NO_x$, and $O_3$.
- The current atmospheric pollution levels in Mae Moh.
- Possible major sources of atmospheric pollution in Mae Moh.

Key premises:

- The WTP would be elicited only for the purpose of estimating the value of a clean atmosphere. While information can be used for environmental communication, it is up to the authorities whether they would actually change policies and act toward the mitigation of pollutants.
- Reporting the WTP would not lead to any obligation for payment. The authorities would remain responsible for securing the budget from various sources.
- The amount being elicited would be the individual-level WTP and not the household-level WTP.

The respondents were first asked whether they would be willing to pay whatever amount of money for the mitigation of atmospheric pollution. Those who chose "not willing to pay any amount" were recorded with zero WTP value and were subjected to a follow-up question to provide their reasons. The respondents were asked to undergo the process of a bidding game conditional on being willing to pay a non-zero amount. For elicitation of WTP values, two hypothetical scenarios for mitigating atmospheric pollution were presented to the respondents: 50% and 80% mitigation in atmospheric pollution in general. A study in China applied the CVM with a set of similar hypothetical scenarios for 30%, 45%, and 60% mitigation of smog in the urban atmosphere [88]. The first-tier question was presented as follows:

- Would you be willing to pay for atmospheric pollution mitigation in Mae Moh? (Yes/No)
- If the respondent answered 'Yes', subsequent questions were asked to elicit the amount of WTP as follows:
- Would you be willing to pay THB __ per month for 50% mitigation in atmospheric pollution in Mae Moh?
- The bid amount started with THB 10 per month. If the answer was 'Yes', then the bid amount was raised by THB 10 until the answer finally became 'No'. The maximum amount for which the answer was 'Yes' was recorded as the individual WTP for atmospheric pollution mitigation by the specified percentage (i.e., 50% or 80%). Given the WTP for the 50% mitigation scenario, the bidding process for the 80% mitigation scenario was initiated. Further, for the 50% mitigation scenario, the elicitation was conducted for the six specific major pollutants as well, namely, $PM_{2.5}$, $PM_{10}$, $SO_2$, $NO_x$, $O_3$, and CO. An example is provided below:
- Would you be willing to pay THB __ per month for 50% mitigation in $PM_{2.5}$ concentrations in Mae Moh?

*3.4. Statistical Analysis*

The analysis of the collected data consisted of three parts. The first part presented the descriptive statistics for the respondents' socio-economic profile, perception of atmospheric pollution including satisfaction with the atmospheric quality and its management, perception of major sources of emissions of pollutants, and the individually elicited WTP values. As for the WTP, both the arithmetic mean and the weighted mean were presented with the weights being the sub-district level sample-population ratios. The second part estimated the aggregate value of the two hypothetical atmospheric quality improvement scenarios for the entire population of the Mae Moh district. The third part analyzed the factors associated with the individual WTP using multiple regression methods, specifically, the probit, bivariate tobit, and double-hurdle methods.

For the aggregation of the WTP, the accumulated 'yes' bids were graphically translated into the demand curve for the respondents in Mae Moh. Demand curves tend to have a downward slope from left to right, representing a negative relationship between the WTP and the number of respondents who can afford the given WTP level (i.e., lower than the individual WTP). The downward slope characterizes the concept of diminishing marginal WTP [87]. By calculating the size of the area under the demand curve, the total WTP was obtained for the sampled respondents. For the calculation of the population-level aggregate value, the weighted mean WTP for the sample was multiplied by the population-sample ratio.

There are various regression methods that can be employed depending on the type of variables and underlying assumptions. The probit regression is used when the dependent variable is in the binary form (i.e., it takes only two values such as adoption and non-adoption) [89,90]. In this study, the probit model is applied to analyze the dependent variable with two outcomes: "willing to pay" and "not willing to pay" for atmospheric pollution mitigation, where the value of 1 is assigned for "willing to pay" and 0 for "not willing to pay". In other words, this binary dependent variable represents the willingness to participate in payment for atmospheric pollution mitigation. Accordingly, the independent variables were the potential factors associated with the willingness to participate. The probit regression model for the two hypothetical mitigation scenarios can be expressed as follows:

$$Y_{ij} = \begin{cases} 1 \; if \; Y_{ij}^* > 0 \\ 0 \; if \; Y_{ij}^* \leq 0 \end{cases} \tag{2}$$

$$Y_{ij}^* = X_j \beta_j + \varepsilon_{ij} \tag{3}$$

$$\varepsilon_{ij} \sim N(0, \; 1) \tag{4}$$

where $Y_{ij}$ is the observed binary variable representing whether the respondent is willing or not willing to pay for 50% ($j = 1$) and 80% ($j = 2$) mitigation of atmospheric pollutants by respondent $i$ ($i = 1, 2, \ldots \ldots , 200$), where the value of 1 represents "willing to pay" and 0 represents "not willing to pay". $Y_{ij}^*$ is the latent variable representing the respondent's likelihood of willing to pay, $X_j$ is the vector of the explanatory variables representing the factors associated with the likelihood of willing to pay for atmospheric pollution mitigation, $\beta_j$ is a vector of the coefficients to be estimated, and $\varepsilon_{ij}$ is the stochastic error term following the standard normal distribution with a mean of 0 and standard deviation of 1. The coefficients $\beta_j$ are estimated by the maximum likelihood estimation (MLE) procedure [89].

Another model used for the analysis is the bivariate tobit regression [91], which is a variation of the tobit regression with two interrelated dependent variables and the same set of independent variables, which can be expressed as follows:

$$R_{ij} = \begin{cases} R_{ij}^* = x_i\gamma_j + \varepsilon_{ij}, \; if \; R_{ij}^* > 0; \; \varepsilon_{ij} \; \sim \; BVN(0, \Sigma) \\ \qquad\qquad 0, \; if \; R_{ij}^* \leq 0 \end{cases} \tag{5}$$

where $R_{ij}$ is the censored dependent variable representing the WTP amount which is censored from the left at zero, where the subscript $j$ represents the two hypothetical mitigation scenarios, $x_i$ represents respondents' socioeconomic characteristics and their perceptions toward atmospheric pollution in Mae Moh, including the unity term, $\gamma_j$ is a vector of parameters to be estimated, including the intercept term, and $\varepsilon_{ij}$ is the error term which is bivariate normally distributed.

Lastly, the double-hurdle model was employed [92,93], which incorporates Heckman's two-step sample selection correction model into the tobit [94,95]. The difference between the double-hurdle and Heckman's is that the latter applies to a truncated dependent variable in the second step, whereas the former applies to a censored dependent variable in the second tier, which contains much more information than a truncated variable. Essentially, the double-hurdle model allows the WTP to result from the two interlinked processes: participation and extent of participation [96], and the second-tier result is corrected for the selection bias when it exists. Using the latent dependent variable, the first tier or the participation equation is expressed as follows:

$$y_{1i}^* = x_i\alpha_1 + u_i \tag{6}$$

where $y_{1i}^*$ is the likelihood that the $i$–th respondent is willing to participate in paying for atmospheric pollution mitigation, $\alpha_1$ is a vector of parameters to be estimated, and $u_i$ is the random error term. Accordingly, the participation ($y_1$) is defined as follows:

$$y_{1i} = \begin{cases} 1 \; if \; y_{1i}^* > 0 \\ 0 \; otherwise \end{cases} \tag{7}$$

For the second tier, the amount that the $i$-th respondent is willing to pay for atmospheric pollution mitigation scenarios is linked to the unobservable latent variable ($y_{2ij}^*$) as follows:

$$y_{2ij}^* = x_i\alpha_{2j} + \sigma_j \cdot IMR_{ij} + v_{ij} \tag{8}$$

$$y_{2ij} = \begin{cases} y_{2ij}^* \; if \; y_{1i}^* > 0 \\ \quad 0 \; otherwise \end{cases} \tag{9}$$

where $y_{2ij}$ represents the extent of the WTP for the hypothetical atmospheric pollutants mitigation scenarios, $\alpha_{2j}$ is a vector of parameters to be estimated, $IMR_{ij}$ is the inverse Mill's ratio constructed from the first tier result as an instrument to control for a selection bias when it exists [96–98], and $v_i$ is the random error term.

Table 4 illustrates the variables included in the regression analyses. The numerical WTP is censored at zero, which corresponds to the cases of zero value in the binary WTP.

A total of sixteen independent variables were analyzed in the analysis which covered the respondents' socioeconomic profile as well as their perception toward atmospheric quality and pollution sources. The ordinal and categorical variables (education and occupation) were converted into sets of dummy variables for inclusion in regressions.

**Table 4.** The dependent and independent variables included in the analysis of factors associated with individual willingness to pay (WTP) for mitigation of atmospheric pollution in Mae Moh.

| Variables | Scale | Description | Expected Sign | Relevant Literature |
|---|---|---|---|---|
| | | Dependent Variables | | |
| Binary WTP (likelihood of WTP) | Binary | 1 if willing to pay some amount for atmospheric pollution reduction, 0 if not willing to pay any amount. | | [99] |
| Numerical WTP (WTP amount) | Ratio Scale | The amount the respondent is willing to pay for atmospheric pollution mitigation. (THB/month) | | [100] |
| | | Independent Variables | | |
| Gender | Binary | Sex of respondent (1 if male, 0 if female). | Positive | [65] |
| Age | Ratio Scale | Age of respondent (years) | Positive | [100] |
| Income | Ratio Scale | Monthly income (THB) | Positive | [64] |
| Expenditure | Ratio Scale | Respondent's monthly expenditure (THB) | Positive | [64] |
| Education | Ordinal (dummy coded in regression) | 1 if no completed school (base group), 2 if primary school, 3 if high school, 4 if vocational school, 5 if university degree | Positive | [100] |
| Occupation | Categorical (dummy coded in regression) | 1 if no job (base group) 2 if employee (government/corporate), 3 if business owner, 4 if farmer, 5 if student or housewife | Positive | [76] |
| Household headship | Binary | 1 if respondent is a household head, 0 otherwise | Positive | [72] |
| Health condition | Binary | 1 if respondent is healthy, 0 otherwise | Negative | [74] |
| Sick from atmospheric pollution | Binary | 1 if sick due to atmospheric pollution, 0 otherwise | Positive | [74] |
| The EGAT as pollution source | Binary | 1 if the EGAT is perceived as a major source, 0 otherwise | Positive | [80] |
| Biomass burning as pollution source | Binary | 1 if biomass burning is perceived as a major source, 0 otherwise | Positive | [101] |
| Transportation as pollution source | Binary | 1 if transportation is perceived as a major source, 0 otherwise | Positive | [9] |
| Household as pollution source | Binary | 1 if household activities are perceived as a major source, 0 otherwise | Positive | [72] |
| Small factories as pollution source | Binary | 1 if small factories are perceived as a major source, 0 otherwise | Positive | [24] |
| Satisfaction with atmospheric quality | Binary | 1 if satisfied with atmospheric quality, 0 otherwise | Negative | [65] |
| Satisfaction with management of atmospheric quality | Binary | 1 if satisfied with management of atmospheric quality by local authorities, 0 otherwise | Negative | [65] |

## 4. Results

### 4.1. Respondents' Socioeconomic Profile

Table 5 presents the descriptive statistics of the respondents' socioeconomic profile. The average age of the respondents was 52 years. The percentage of male respondents exceeded females by 9%. Around 60% of the respondents labeled themselves as household heads. In terms of educational attainment, more than half of the respondents had primary

education or below, while only 14% finished vocational or university level of education. The majority of the respondents were either business owners (33%) or farmers (46%). One-tenth of the respondents were students and housewives. The average annual income was THB 96,180 per annum (USD 3232), while the average annual expenditure was THB 111,372 per annum (USD 3742). This suggests that many households in Mae Moh needed to borrow money from external sources in order to survive, which might lead to debt accumulation over time. In relation to this, the amount of savings was reported on the ordinal scale, indicating that a staggering 50% of the respondents had no savings, and only 4% claimed high savings.

**Table 5.** Respondents' socioeconomic profile (*n* = 200).

| Variable | Mean | Std. Dev. | Min | Max |
|---|---|---|---|---|
| Age (years) | 52.04 | 11.29 | 5 | 90 |
| Income (THB/annum) | 96,174 | 44,101 | 0 | 216,000 |
| Expense (THB/annum) | 116,232 | 96,770 | 12,000 | 1080,036 |
| Net income (THB/annum) | −20,058 | 90,535 | −984,036 | 168,000 |

| Variable | Frequency | Percentage |
|---|---|---|
| Gender | | |
| Female | 91 | 45.5 |
| Male | 109 | 54.5 |
| Household head | | |
| No | 79 | 39.5 |
| Yes | 121 | 60.5 |
| Education | | |
| None | 17 | 8.5 |
| Primary | 91 | 45.5 |
| High school | 64 | 32.0 |
| Vocational | 14 | 7.0 |
| University | 14 | 7.0 |
| Savings | | |
| None | 100 | 50.0 |
| Low | 35 | 17.5 |
| Medium | 58 | 29.0 |
| High | 7 | 3.5 |
| Ocupation | | |
| No job | 16 | 8.0 |
| Employee | 7 | 3.5 |
| Business owner | 65 | 32.5 |
| Farmer | 92 | 46.0 |
| Student and housewife | 20 | 10.0 |
| Satisfaction with atmospheric quality | | |
| Satisfied | 163 | 81.5 |
| Not satisfied | 37 | 18.5 |
| Satisfaction with management of atmospheric quality | | |
| Satisfied | 182 | 91.0 |
| Not satisfied | 18 | 9.0 |

Note: USD 1 = THB 29.76 as of 31 December 2019.

In terms of satisfaction with atmospheric quality in Mae Moh, 82% reported satisfaction while 18% reported dissatisfaction. In terms of satisfaction with atmospheric quality management, 91% reported satisfaction while 9% reported dissatisfaction. These results suggest that locals were generally satisfied with atmospheric quality in Mae Moh nowadays.

Table 6 shows the perceived major sources of atmospheric pollution in Mae Moh, where multiple choices were allowed. A total of 70% of the respondents perceived the EGAT as a major source of atmospheric pollution, while 41% perceived biomass burning as a major source. Small factories appeared to be the least concern among residents in Mae Moh.

**Table 6.** Perceived major sources of atmospheric pollution in Mae Moh (*n* = 200).

| Major Source | Answer | Frequency | Percentage |
|---|---|---|---|
| The EGAT | Yes | 140 | 70.0 |
| | No | 60 | 30.0 |
| Biomass Burning | Yes | 82 | 41.0 |
| | No | 118 | 59.0 |
| Household | Yes | 45 | 22.5 |
| | No | 155 | 77.5 |
| Transportation | Yes | 39 | 19.5 |
| | No | 161 | 80.5 |
| Small Factories | Yes | 10 | 5.0 |
| | No | 190 | 95.0 |

*4.2. Individual Willingness to Pay*

Table 7 shows the frequency distribution of the individual WTP per month for 50% and 80% mitigation of atmospheric pollution, respectively. Over two-thirds (68%) of the respondents were not willing to pay any amount for mitigation by either 50% or 80%, while 32% of the respondents showed a certain amount of WTP. Some contrast was noted in the distribution of WTP between the 50% and 80% mitigation scenarios. The proportion of the respondents with the WTP being THB 100 or higher was considerably higher for the 80% mitigation scenario. Around 13% of the respondents were willing to pay THB 100 or higher for the 80% mitigation, while only 8.5% were willing to pay THB 100 or higher for the 50% mitigation scenario.

Table 8 summarizes the descriptive statistics of the individual WTP for 50% and 80% mitigation of atmospheric pollutants for each sub-district. As expected, the arithmetic mean of WTP for the 80% mitigation scenario was higher than that for the 50% mitigation scenario across the board. The median was zero because the majority of respondents were not willing to pay. The standard deviation was generally higher for the 80% mitigation scenario. The Mae Moh Sub-district exhibited the highest mean WTP, while the sub-district with the lowest mean WTP was Ban Dong and Na Sak for the 50% mitigation and Chang Nuea for the 80% mitigation.

Table 9 presents the descriptive summary of the individual WTP for mitigation of atmospheric pollutants under different hypothetical scenarios. The arithmetic mean WTP for mitigation of overall pollutants was THB 18.7 and 22.8 per month for the 50% and 80% mitigation scenarios, respectively, which translate into the annual values of THB 224.4 and 273.6. The weighted mean was THB 20.94 and 25.66 per month (THB 251.3 and 307.9 per annum). The median WTP was zero because the majority of respondents were not willing to pay. The result confirms that local residents attached higher values to the 80% mitigation scenario. In addition, the WTP for 50% mitigation of the specific major pollutants was obtained. The weighted WTP for mitigation of $PM_{2.5}$ and $PM_{10}$ was THB 19 and 4.1 per month, respectively (THB 228 and 49.2 per annum). On the other hand, the weighted mean WTP for the other four major pollutants was found to be negligibly low. The result suggests that the pollutants of major concern to local residents today were $PM_{2.5}$ and $PM_{10}$.

**Table 7.** Frequency distribution of the bid amount for the willingness to pay (WTP) for mitigation of atmospheric pollution under two hypothetical scenarios.

| WTP (THB/Month) | 50% Mitigation Hypothetical Scenario | | 80% Mitigation Hypothetical Scenario | |
|---|---|---|---|---|
| | Num. of Respondents | Percentage | Num. of Respondents | Percentage |
| 0 | 136 | 68.0 | 136 | 68.0 |
| 10 | 6 | 3.0 | 4 | 2.0 |
| 20 | 7 | 3.5 | 5 | 2.5 |
| 30 | 4 | 2.0 | 3 | 1.5 |
| 40 | 6 | 3.0 | 6 | 3.0 |
| 50 | 15 | 7.5 | 10 | 5.0 |
| 60 | 7 | 7.0 | 8 | 4.0 |
| 70 | 0 | 0 | 0 | 0 |
| 80 | 2 | 1.0 | 2 | 1.0 |
| 90 | 0 | 0 | 0 | 0 |
| 100 | 14 | 7.0 | 21 | 10.5 |
| 110 | 1 | 0.5 | 1 | 0.5 |
| 120 | 0 | 0 | 0 | 0 |
| 130 | 0 | 0 | 0 | 0 |
| 140 | 1 | 0.5 | 1 | 0.5 |
| 150 | 0 | 0 | 0 | 0 |
| 160 | 0 | 0 | 0 | 0 |
| 170 | 0 | 0 | 0 | 0 |
| 180 | 0 | 0 | 0 | 0 |
| 190 | 0 | 0 | 0 | 0 |
| 200 | 1 | 0.5 | 3 | 1.5 |

Note: USD 1 = THB 29.76 as of 31st December 2019.

**Table 8.** Individual willingness to pay (WTP) (THB per month) for mitigation of atmospheric pollutants under the two hypothetical mitigation scenarios for each sub-district ($n = 200$).

| Sub-District | WTP for 50% Mitigation Scenario | | | WTP for 80% Mitigation Scenario | | |
|---|---|---|---|---|---|---|
| | Mean | Median | SD | Mean | Median | SD |
| Bang Dong | 15.75 | 0 | 25.30 | 24.50 | 0 | 43.97 |
| Na Sak | 15.75 | 0 | 27.54 | 18.00 | 0 | 30.90 |
| Chang Nuea | 16.00 | 0 | 35.65 | 16.00 | 0 | 35.00 |
| Mae Moh | 27.25 | 0 | 44.26 | 34.25 | 0 | 54.01 |
| Sop Pat | 18.75 | 0 | 35.60 | 21.50 | 0 | 38.06 |

**Table 9.** Descriptive statistics of individual willingness to pay (WTP) (THB per month) for mitigation of atmospheric pollutants under different hypothetical scenarios ($n = 200$).

| Hypothetical Mitigation Scenario | Arithmetic Mean | Weighted Mean * | Median | Standard Deviation |
|---|---|---|---|---|
| Overall; 50% | 18.70 | 20.94 | 0 | 34.28 |
| Overall; 80% | 22.80 | 25.66 | 0 | 41.27 |
| $PM_{2.5}$; 50% | 17.25 | 19.01 | 0 | 34.29 |
| $PM_{10}$; 50% | 3.95 | 4.08 | 0 | 11.56 |
| $SO_2$; 50% | 0.50 | 0.41 | 0 | 7.07 |
| $NO_x$; 50% | 0.50 | 0.41 | 0 | 7.07 |
| $O_3$; 50% | 0.50 | 0.41 | 0 | 7.07 |
| CO; 50% | 1.00 | 0.75 | 0 | 9.97 |

* Weighted by the subdistrict-level representation ratio in Table 3. The Wilcoxon signed-rank test showed a significant difference in the WTP for overall mitigation between the two hypothetical scenarios ($p = 0.002$). USD 1 = THB 29.76 as of 31 December 2019.

### 4.3. Aggregate Willingness to Pay

Figure 3 plots the individual WTP values and the corresponding numbers of respondents whose WTP was higher than the given WTP value. In other words, Figure 3 represents sample-level demand curves. The plots for both scenarios exhibited downward sloping curves, which is in line with the theoretically predicted diminishing marginal WTP [102]. Furthermore, the revealed non-linearity of the demand curves indicates that a uniform change in the hypothetical payment would lead to a non-uniform change in demand for improved atmospheric quality. It was confirmed that the demand for the 80% mitigation scenario was consistently greater than that for the 50% mitigation scenario at all the WTP levels. Accordingly, the total area under the curve (i.e., the total WTP) for 80% mitigation was larger than that for 50% mitigation, indicating that a 30%-point further reduction in the pollution would indeed accrue higher values to residents in Mae Moh.

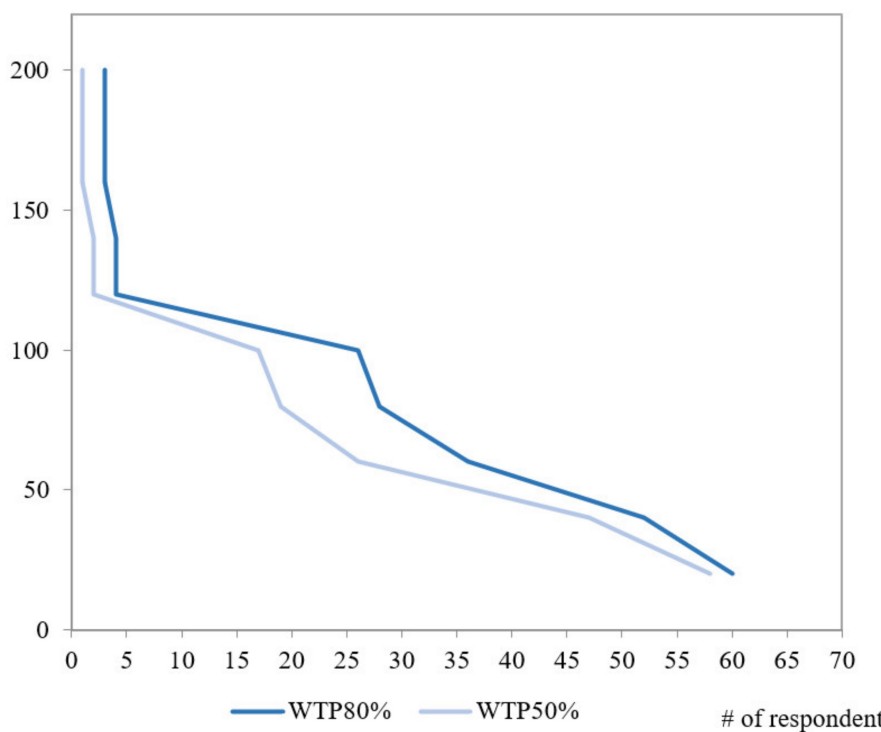

**Figure 3.** Relationship between the individual willingness to pay (WTP/month) and the number of respondents for the 50% and 80% atmospheric pollution mitigation scenarios.

To estimate the aggregate economic value of the two hypothetical mitigation scenarios, the weighted mean WTP was multiplied by the population–sample ratio. Plugging THB 20.94 and 25.66 into the weighted mean WTP for the 50% and 80% mitigation scenarios, respectively, and 39,831/200 = 199.155 into the population-sample ratio, as well as annualizing the values, the aggregate economic value was computed as THB 10,008,734 (USD 336,294) and THB 12,264,762 (USD 412,096) per annum, respectively.

### 4.4. Individual Factors Associated with the Willingness to Pay

Table 10 presents the results of the probit regression on factors associated with the likelihood of willingness to participate in payment for atmospheric pollution mitigation, where the marginal effects on the likelihood were shown in addition to the coefficients. The factors that appeared to have significant associations with the likelihood of participation were education (primary and university levels), income, expense, occupation (business owner and farmer), satisfaction with atmospheric quality, and the EGAT as a perceived

source of pollution. Those who attended up to primary school and university had, on average, 41.8% points and 79.6% points higher likelihood of participation in payment for atmospheric pollution mitigation than those with no formal education attainment (i.e., the reference category), holding other variables unchanged. Income and expense had positive and negative effects on the probability of willing to participate, respectively. More specifically, a one-thousand THB increase in income increased the probability of willing to participate by 13% points, while an increase in expenses by one thousand THB decreased the probability of willing to participate by 9.6% points, on average, holding other variables constant. In terms of occupation, business owners and farmers had 21.8% points and 28.3% points lower likelihood of willing to participate, in comparison with those with no job. Satisfaction with current atmospheric quality showed negative effects (−25.8% points) on the likelihood of willing to participate, indicating that those satisfied with the current atmospheric quality in Mae Moh attached less value to further mitigation of pollutants. As for perceived major sources of pollution, the EGAT was the only source associated with the likelihood of willing to participate in payment, whereby those who perceived the EGAT as a major source were 14.6% points more likely to be willing to participate.

**Table 10.** Factors associated with the probability of willing to participate in payment for mitigation of atmospheric pollution: the probit regression.

| Variable | Coefficient (SE) | Marginal Effect |
|---|---|---|
| Age (years) | 0.001 (0.082) | 0.000 |
| Age squared (years$^2$) | 0.000 (0.001) | 0.000 |
| Gender (1 if male, 0 if female) | −0.560 (0.420) | −0.125 |
| *Education (base = no education)* | | |
| Primary | 1.772 * (0.997) | 0.418 |
| High school | 1.414 (1.078) | 0.383 |
| Vocational | 0.745 (1.509) | 0.217 |
| University | 2.541 * (1.380) | 0.796 |
| Household Head | −0.034 (0.442) | −0.007 |
| Income (in thousand THB) | 0.598 *** (0.121) | 0.130 |
| Expense (in thousand THB) | −0.443 *** (0.099) | −0.096 |
| *Occupation dummies (base = no job)* | | |
| Employee (govt. and private sector) | −0.471 (0.762) | −0.096 |
| Student and housewife | 0.970 (1.347) | 0.307 |
| Business owner | −1.265 * (0.662) | −0.218 |
| Farmer | −1.356 * (0.645) | −0.283 |
| Savings (1 if there is, 0 if no saving) | −0.021 (0.378) | 0.004 |
| Satisfaction with atmospheric quality (1 if satisfied, 0 otherwise) | −0.914 ** (0.427) | −0.258 |
| Satisfaction with management of atmospheric quality (1 if satisfied, 0 otherwise) | 0.112 (0.772) | 0.023 |
| Health (1 if healthy, 0 otherwise) | −0.213 (0.527) | −0.046 |
| Sickness from pollution (1 if sick, 0 otherwise) | 0.232 (0.352) | 0.050 |
| *Perceived sources of pollution* | | |
| The EGAT (1 if perceived as a source, 0 otherwise) | 0.802 * (0.413) | 0.146 |
| Biomass/open burning (ditto) | 0.370 (0.432) | 0.083 |
| Transportation (ditto) | 0.277 (0.532) | 0.066 |
| Household (ditto) | −0.467 (0.462) | −0.087 |
| Small factories (ditto) | −0.510 (1.178) | −0.66 |
| Constant | −1.899 (2.293) | 0.321 |

Dependent variable: willing to participate in payment for mitigation of atmospheric pollution (1 if willing to participate, 0 otherwise)
$n$ = 200; Likelihood Ratio $\chi^2(24)$ = 146.362 ($p$ = 0.000);
Log likelihood = −51.678; McFadden's Pseudo $R^2$ = 0.586

Note: *p*-values are in parentheses. ***, **, and * indicate $p < 0.01$, $< 0.05$, and $< 0.10$, respectively.

Table 11 presents the results from the bivariate tobit and double-hurdle models. For the double-hurdle model, only the second-tier results were shown as the first-tier was identical to the probit results in Table 10. The two models yielded somewhat consistent results though with some contrast, whilst the two hypothetical extents of mitigation (50% and 80%) yielded some differences in associated factors. Another overall observation is that

the two tiers of the double-hurdle results were in contrast with each other as they revealed largely different sets of significant factors. This implies that the hypothetical participation in payment and the hypothetical extent of payment were determined by different processes, supporting the validity of the use of the double-hurdle model with our valuation data. It also suggests that the double-hurdle results may be more reliable than the bivariate tobit, though it remains wise to triangulate the results.

**Table 11.** Individual factors associated with the values of individual willingness to pay (WTP) for atmospheric pollution mitigation in Mae Moh: the bivariate tobit and the second tier of the double hurdle regressions (*n* = 200).

| Variable | Extent of Hypothetical Mitigation of Atmospheric Pollution | | | |
|---|---|---|---|---|
| | 50% Mitigation | | 80% Mitigation | |
| | Bivariate Tobit (SE) | Double-Hurdle (SE) | Bivariate Tobit (SE) | Double-Hurdle (SE) |
| Age | −0.775 (0.812) | −6.113 * (3.083) | −2.116 ** (1.021) | −13.535 *** (3.124) |
| Age squared | 0.006 (0.008) | 0.060 * (0.035) | 0.015 (0.010) | 0.114 *** (0.033) |
| Gender | −3.407 (5.705) | 4.813 (14.538) | −3.135 (7.716) | −3.068 (13.966) |
| *Education (base = no education)* | | | | |
| Primary | −0.107 (8.885) | −12.790 (55.506) | −6.457 (11.176) | 38.649 (51.999) |
| High school | 0.968 (9.828) | 1.522 (56.549) | 4.325 (12.362) | 27.740 (53.034) |
| Vocational | 10.361 (13.017) | 33.987 (58.117) | 5.810 (16.374) | 30.579 (55.213) |
| University | −1.185 (12.177) | 9.713 (60.112) | 2.716 (15.317) | 33.925 (56.747) |
| Household head | −2.732 (6.126) | 0.203 (12.661) | 2.206 (7.706) | 23.475 (12.224) |
| Income (in thousand) | 5.568 *** (0.776) | 3.627 * (2.105) | 5.733 *** (0.976) | 3.356 (2.099) |
| Expense (in thousand) | −2.620 *** (0.541) | −1.321 (2.128) | −3.067 *** (0.680) | −2.161 (2.072) |
| *Occupation dummies (base = no job)* | | | | |
| Government and/or corporate employees | −5.483 (14.092) | 43.526 (34.447) | 0.342 (17.726) | 44.077 (32.486) |
| Student and housewife | −7.909 (14.217) | −12.283 (38.366) | −21.399 (17.884) | −47.777 (37.397) |
| Business owner | 1.546 (7.220) | 56.251 *** (22.321) | −9.117 (9.082) | 23.238 (20.150) |
| Farmer | 2.786 (6.722) | 67.810 *** (20.883) | −4.969 (8.455) | 52.152 *** (18.148) |
| Savings (1 if yes, 0 if no) | −1.440 (5.176) | −19.556 (13.589) | 0.678 (6.511) | −20.690 (12.980) |
| Satisfaction with atmospheric quality (1 if yes, 0 if no) | 1.291 (5.983) | 32.036 *** (12.544) | 2.993 (7.525) | 43.108 *** (12.391) |
| Satisfaction with management of atmospheric quality (1 satisfied, 0 otherwise) | 4.499 (8.575) | −4.550 (17.414) | −0.939 (10.787) | −15.107 (16.686) |
| Health (1 if healthy, 0 otherwise) | 3.295 (0.622) | 5.217 (20.851) | 6.117 (8.396) | 9.507 (21.197) |
| Pollution sickness (1 if sick, 0 otherwise) | 0.583 (5.435) | −0.453 (15.203) | −3.374 (0.622) | −26.602 (15.231) |
| *Pollution sources (1 if perceived as a source, 0 otherwise)* | | | | |
| The EGAT | 18.903 *** (4.824) | 18.546 (14.092) | 24.483 *** (6.068) | 22.567 * (13.864) |
| Biomass/open burning | 18.814 *** (4.978) | 36.311 *** (12.825) | 13.264 ** (6.261) | 18.350 (12.629) |
| Transportation | −0.238 (5.531) | −22.492 (13.942) | −10.583 (0.303) | −8.544 (13.780) |
| Household | −1.661 (5.721) | 10.376 (13.287) | −7.237 (0.461) | 4.397 (13.653) |
| Small factories | −18.315 * (9.973) | −4.161 (30.782) | −26.415 ** (12.544) | −14.995 (32.113) |
| Constant | −4.503 (23.819) | 84.156 (88.903) | 47.218 ** (29.961) | 320.586 *** (86.639) |
| | Wald χ² (24) = 146.13 Log likelihood = −1808.24 *p* = 0.000 | Wald χ² (24) = 43.17 Log Likelihood = −338.96 *p* = 0.010 | Wald χ² (24) = 146.13 Log likelihood = −1808.24 *p* = 0.000 | Wald χ² (24) = 43.07 Log likelihood = −344.54 *p* = 0.001 |

Dependent variable: willingness to pay (WTP) for 50% and 80% mitigation of atmospheric pollution in Mae Moh (THB/annum). USD 1 = THB 29.76 as of 31 December 2019. *p*-values are in parentheses. ***, **, and * indicate <0.01, <0.05, and <0.10, respectively.

With regard to the specific factors, age (and age squared), income, expenses, occupation (business owner and farmer), satisfaction with atmospheric quality, and perceived major source of pollution (the EGAT, biomass burning, and small factories) were identified as significant individual factors associated with the WTP values. Age exhibited non-linear effects on the WTP in both mitigation scenarios, as per the double-hurdle results. Applying the differential calculus to the set of estimated coefficients indicated that the predicted WTP was minimized at the age of 72 and 63 years old for the double-hurdle model under 50% and 80% mitigation scenarios, respectively. This means that the effects of age were negative among the younger population and nearly zero among the senior population, suggesting that the younger population was willing to pay higher amounts for atmospheric population mitigation.

The coefficients of the double-hurdle model for income and expenses for both scenarios were positive and negative, respectively, indicating that higher income increases the WTP, while higher expenses decrease the WTP for atmospheric pollution mitigation. On average for the 50% mitigation scenario, an increase in monthly income by THB 1000 leads to an increase in WTP by THB 5.6 per month, while an increase in monthly expenses by THB 1000 leads to a decrease in WTP by THB 2.6 per month. For the 80% mitigation scenario, a THB 1000 increase in monthly income leads to an increase in WTP by THB 5.7 per month, while a THB 1000 increase in monthly expenses leads to a decrease in WTP by THB 3.1 per month. This result makes logical sense as individuals with higher income (expenses) should be more (less) able to afford the higher payment. The double-hurdle model revealed that business owners and farmers were more willing to pay higher amounts for the 50% mitigation scenario, compared to those with no job, where the average difference was THB 56.2 and 67.8 per month (THB 674.4 and 813.6 per annum) respectively, holding covariates unchanged.

Satisfaction with the current atmospheric quality was another positive factor for both mitigation scenarios. On average, those satisfied with the atmospheric quality in Mae Moh were willing to pay amounts higher by THB 32.0 and 43.1 per month (THB 384.0 and 517.2 per annum) for the 50% and 80% mitigation scenarios, respectively, compared to those unsatisfied. Regarding the perceived major sources of atmospheric pollution, perceiving the EGAT and biomass open burning as major sources led to higher WTP. As per the bivariate tobit, those who perceived the EGAT and biomass burning as major sources were willing to pay amounts higher by THB 18.9 and 18.8 per month (THB 226.8 and 225.6 per annum), respectively, for the 50% mitigation scenario, compared to those who did not perceive the said sources. Likewise, for the 80% mitigation scenario, the marginal effects for the same were THB 24.48 and 13.26 per month (THB 293.8 and 159.1 per annum). Biomass burning in this context refers to various sources including deforestation, forest fires, shifting cultivation, burning of biomass as fuels, as well as agricultural residues [101]. Albeit not shown by the double-hurdle mode, the bivariate tobit pointed to the negative effects of small factories as a perceived source. If the effects exited, those with the perception would be willing to pay amounts lower by THB 18.3 and 26.4 per month (THB 219.6 and 316.8 per annum) on average for the 50% and 80% mitigation scenarios, respectively.

## 5. Discussions

On average, individuals in the Mae Moh district were willing to pay THB 224.4 and 273.6 per person per annum (USD 7.5 and 9.2 per annum) for the 50% and 80% mitigation scenarios, respectively. The mean WTP was remarkably lower compared to the results in the past studies. For example, Filippini and Martínez-Cruz (2016) [103] found, in Mexico, that on average, people were willing to pay about USD 262 per annum for a reduction in atmospheric pollution. Carlsson and Johansson-Stenman (2000) [101] found that the mean WTP for 50% reduction in harmful atmospheric contaminants in Sweden was USD 227 per annum. Wang (2006) [64] found that the mean WTP for a 50% reduction in harmful atmospheric pollutants in Beijing was CYN 143 (USD 17.9) per household per year. Mohammed (2003) [104] estimated the mean WTP for a 50% reduction in atmospheric

pollution caused by road traffic in Rabat-Salé, Morocco, to be USD 96 per capita per annum. The proportion of locals willing to pay for atmospheric pollution mitigation in Mae Moh was also relatively low (32%) compared to other studies such as Liu et al., (2018) [74] in China (53%). The distinct gap in WTP levels may be attributed to the relatively clean air in today's Mae Moh compared to severely polluted major metropolitan cities, as well as the relatively low average income of local citizens in Mae Moh (THB 96,174 or USD 3231 per annum) compared to those other locations.

In terms of the specific pollutants, the average WTP for $PM_{2.5}$ and $PM_{10}$ was THB 17.25 and 3.95 per month (USD 0.58 and 0.13 per month), while the average WTP for other pollutants was negligibly low (less than THB 1.0 per month). The installation of the FGD devices by the EGAT had largely eliminated atmospheric $SO_2$ in Mae Moh [105]. Yet, according to Greenpeace [22], Mae Moh has the highest average annual concentrations of $PM_{2.5}$ and $PM_{10}$ in Thailand. The PCD [84] also reports much higher concentrations of $PM_{2.5}$ and $PM_{10}$ in Mae Moh compared to the other pollutants. It is likely not a coincidence that local residents were willing to pay much higher amounts for PM mitigation than for other pollutants. The literature shows that lasting exposure to PM has negative long-term impacts on human health such as premature mortality and a wide range of morbidity outcomes including respiratory illnesses and cardiovascular diseases [101].

The aggregate benefit of atmospheric pollution mitigation for the Mae Moh District population was THB 10,008,734 and 12,264,762 (USD 336,294 and 412,096) per annum for the 50% and 80% mitigation scenarios, respectively. Compared to previous studies, our aggregate WTP result appears to be lower, which is mainly due to the relatively small population in the Mae Moh District vis-à-vis major cities. For example, Wang (2006) [64] found that the aggregate WTP for a 50% reduction in harmful atmospheric pollutants in Beijing was CYN 336 million (USD 42.1 million) per year. Likewise, Belhaj (2003) [104] found that the aggregate benefit for a 50% reduction in atmospheric pollution caused by road traffic in Rabat-Salé, Morocco, was USD 57 million per year as per the iterative choice method and USD 59 million as per the simplified parametric approach.

According to the law of diminishing marginal utility, as one consumes more and more of a good or service, the total utility increases but at a decreasing rate per additional unit (60). The concept applies to the valuation atmospheric quality improvement as well. Projection using various non-linear functions (i.e., logarithmic function, Cobb–Douglas function, and hyperbolic function) that fit the law suggests that further mitigation from 80% to 100% would not make a significant increment in the WTP. This, combined with the law of increasing marginal abatement costs [105], would suggest that the 50% mitigation scenario could be a more viable option for the authorities.

The majority of local residents perceived the EGAT as the main source of emissions, which is consistent with the ADB report showing that the Mae Moh power station, including the coal mine, had emitted atmospheric pollutants, causing public health concerns to the local population [79]. In the 1990s, $SO_2$ was the main pollutant in the atmosphere, resulting in the lawsuit in 2004 and the eventual ruling in 2015 by the Supreme Administrative Court ordering the EGAT to compensate for the sufferings and losses endured by the locals [106]. In contrast, our mean WTP estimate for each major pollutant today implies that the current residents are mainly concerned about the particulate matter only. Over the past decades, the EGAT took several steps to reduce emissions of harmful pollutants, especially $SO_2$. In 1998, the Flue Gas Desulfurization (FGD) system was installed in Power Plants 4–13, which effectively reduced $SO_2$ emissions by 95% [107]. Moreover, the Continuous Emission Monitoring System (CEMS) was installed in the FGD system for real-time monitoring. According to the law, the $SO_2$ released from the stack must not exceed 320 ppm [107]. Further, 11 atmospheric quality monitoring stations were installed in Mae Moh, in addition to the three ongoing monitoring stations under the responsibility of the Pollution Control Department.

Biomass burning was perceived as another major source of emissions, which is in line with existing studies. Vichit-Vadakan (2011) [24] reported that the three major sources

of atmospheric pollution in Thailand were vehicle emissions, biomass burning, and concentrated industrial zones. Arunrat (2018) [101] found that atmospheric pollution in the northern region of Thailand was mainly due to slash-and-burn practices in highland agriculture where the majority of farmers burned maize crop residues during April and May, mostly in the afternoon hours. Khamkaew et al., (2016) [108] confirmed that biomass burning was a major source of $PM_{2.5}$ pollution in the northern city of Chiang Mai. These studies suggest that biomass burning is one of the major sources of atmospheric pollution in northern Thailand in general, of which Mae Moh may not be an exception, given our findings.

The set of regression analyses identified several factors associated with the likelihood (first tier) and the extent (second tier) of WTP, where different sets of factors were significant between the two tiers, justifying the use of this model. Primary and university level education showed positive effects on the likelihood of willing to pay. While the insignificant effects of high school and vocational school education may be puzzling, it generally makes sense that educated people had a higher likelihood of participation in payment. A study in Nanchang, China, also noted that education was positively associated with the likelihood of WTP among manufacturing workers [74]. Furthermore, our results revealed that being business owners and farmers lowered the likelihood of participation in payment but raised the extent of WTP. This implies that business owners and farmers had polarized valuation, i.e., many of them were unwilling to pay but those willing were willing to pay higher compared to those with no job. Satisfaction with the current atmospheric quality also exhibited the same tendency, i.e., it lowered the likelihood but raised the extent of WTP. Akhtar (2017) [76] found in Lahore, Pakistan, that unfavorable perception of atmospheric quality led to an increase in both the likelihood and extent of the WTP, which contrasts with the polarization observed in our result.

Age showed quadratic effects on the extent of the WTP but showed no significant effect on the likelihood of WTP. The negative marginal effects implied that younger generations had higher WTP. The demographic pyramid of Thailand is shifting toward an aging society, as in some developed countries in Asia and Europe. Since the elderly population is willing to pay less, the mean and aggregate WTP for atmospheric pollution mitigation in Thailand are expected to decline in the future.

Regarding the effects of perceived major sources of atmospheric pollution, those who perceived the EGAT as the major source of emissions had a higher likelihood and extent of WTP than those who did not. Likewise, those who perceived biomass burning, primarily for agricultural purposes, as a major source of emissions tended to have higher WTP. In contrast, those who perceived small factories as a major source tended to have lower WTP. The result may be a mere reflection of the linkage between the perceived extent of pollution and the significance of each source as an emitter.

## 6. Conclusions

Atmospheric pollution has been increasingly severe, adversely affecting human health, environments, the economy, and society at large. This study conducted the economic valuation of atmospheric quality improvement in the Mae Moh District, northern Thailand by quantitatively eliciting local residents' demand for mitigation of atmospheric pollution using the CVM and analyzing the individual factors associated with the WTP using the probit, bivariate tobit, and double-hurdle regression techniques.

The estimated values of 50% and 80% hypothetical mitigation of atmospheric pollution were THB 224.4 and 273.6 (USD 7.5 and 9.2) per annum on average per capita and THB 10,008,734 and 12,264,762 (USD 336,294 and 412,096) per annum on aggregate, respectively. These values may not be remarkably high compared to other cities notorious for atmospheric pollution presumably because the residents in Mae Moh were no longer concerned about the contamination with $SO_2$ which once caused a series of environmental litigation cases. Yet, one-third of the local population remained concerned about $PM_{2.5}$ and $PM_{10}$ as major pollutants today, which might cause serious respiratory symptoms as well

as low visibility of the landscape. The microeconomic theories and the estimated values suggest that the 50% mitigation option may be more viable.

Age effects on the WTP were quadratic where the predicted minimum was in the range of senior ages. The finding indicates that the values of atmospheric quality accruing to the local population may decrease in light of the population dynamics. It may be easier to justify PM mitigation policies and programs sooner rather than later. At the same time, however, income showed positive effects on the likelihood and extent of the WTP. Hence, the higher income expected of future populations may help justify mitigation projects. At the same time, the relatively low average WTP especially for $SO_2$ implies that the FGD based mitigation measures have been successful to date. Yet, the result for PM alludes to the remaining and persistent concern over PM pollution, which allegedly poses a risk of developing respiratory symptoms in the long run, if not in the short run [108]. Although mitigation measures for PM are desired, a massive budget cannot be justified from our result unless further evidence arises in support for it. Furthermore, the relatively low proportion of locals willing to pay for mitigation indicates that the government needs to spearhead the mitigation project in order to effectively mitigate atmospheric pollution in Mae Moh. It would be desirable if the authorities as well as researchers further investigated emissions from the power plant, lignite mine, and open biomass burning to quantify the respective contributions to the atmospheric concentrations of $PM_{2.5}$ and $PM_{10}$.

Lastly, there are certain limitations in this study. First, the CVM questionnaire applied the bidding game format, which is prone to the starting point bias. Triangulation with alternative elicitation methods would help interpret our findings. Second, the individual factors analyzed in this study were the respondents' socioeconomic variables. While this is useful in many ways, investigation and analysis of behavioral constructs may help unveil the mechanism driving the valuation. Third, the survey participants were residents of the Mae Moh District and did not include other nearby districts, where residents could potentially be suffering from the pollutants drifting from Mae Moh. Fourth, this study applied stratified quota sampling, which was not purely probabilistic. Although the weighted calculation was utilized to minimize the potential selection bias, further research with a larger and representative sample may enhance the reliability of the findings. Fifth, individual income reported by the respondents might be understated on grounds that the average bottom line was negative. A more nuanced elicitation of income with an itemized breakdown may help improve the accuracy of the variable.

**Author Contributions:** Conceptualization, W.S. and T.W.T.; data curation, W.S., formal analysis, W.S. and T.W.T.; funding acquisition, T.W.T. and N.S.; investigation, W.S., T.W.T., E.W. and N.S.; methodology, W.S. and T.W.T.; project administration, W.S.; resources, W.S., T.W.T., E.W. and N.S.; software, T.W.T.; supervision, T.W.T., E.W. and N.S.; validation, W.S., T.W.T., E.W. and N.S.; visualization, T.W.T.; writing—original draft, W.S.; writing—review and editing T.W.T. All authors have read and agreed to the published version of the manuscript.

**Funding:** The authors acknowledge funding from the UK Research and Innovation's Global Challenges Research Fund (UKRI GCRF) through the Trade, Development and the Environment Hub project (project number ES/S008160/1) led by the UN Environment Programme World Conservation Monitoring Centre (UNEP-WCMC). The fieldwork was funded by the research grant as part of the Asian Institute of Technology Fellowship. The study was based on the first author's dissertation research which was supported by the Royal Thai Government Fellowship.

**Institutional Review Board Statement:** Not applicable.

**Informed Consent Statement:** Not applicable.

**Data Availability Statement:** The primary dataset collected and analyzed for this study will be provided upon request.

**Acknowledgments:** The authors express their gratitude to Bunyanit Wongrukmit, the Governor of the EGAT, for arranging EGAT officers to render assistance in gathering the information and data needed for this research. The authors also acknowledge Noppadol and Paingduan (EGAT) for

providing the requested information and data and for their cooperation and support during the field work. The authors are thankful to the survey respondents who agreed to provide valuable information during the interviews.

**Conflicts of Interest:** The authors declare no conflict of interest. The funder had no role in the design of the study; in the collection, analyses, or interpretation of data; in the writing of the manuscript, or in the decision to publish the results.

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
