# Peer review of "Valuation of Local Demand for Improved Air Quality: The Case of the Mae Moh Coal Mine Site in Thailand"

_atmosphere, doi:10.3390/atmos12091132_

Round 1

Reviewer 1 Report

The publication is prepared in a methodical manner at a good level.

It is true that there are doubts about the size of the sample and the nature of the research should be considered very preliminary, but the authors raised a very important issue for the development and implementation of a strategy related to air quality and creating a policy related to air protection, which is why I positively assess and recommend the text for publication.

One remark: please include in the text of the publication information about the date of the conducted research. 

Author Response

Thank you very much for the thoughtful comments.
Below we provide the rejoinder.

Comment 1-1

The publication is prepared in a methodical manner at a good level. It is true that there are doubts about the size of the sample and the nature of the research should be considered very preliminary, but the authors raised a very important issue for the development and implementation of a strategy related to air quality and creating a policy related to air protection, which is why I positively assess and recommend the text for publication.

Response 1-1

Thank you very much for taking the time to review our paper and providing favorable feedback.

Comment 1-2

One remark: please include in the text of the publication information about the date of the conducted research.

Response 1-2

Thank you so much for the useful comment. Accordingly, we have included the dates of the field work at the beginning of Section 3.3.

Best wishes,

Reviewer 2 Report

This study assessed local demand for better air quality in the Mae Moh, Thailand by conducting face-to-face interview and statistical analysis on it. 

While the interview result can give valuable results, a clear objective is lacking in logical reasoning. That is, the study found WTP from 32% of the respondents and analyzed factors that can characterize respondents showing WTP but did not gives its meaning that can be applied to policy making.

I would suggest clarify an objective and reorganize the manuscript. Since the interview results are valuable, it can turn into an important paper. 

General comments:

I suggest to clarify a goal of the study. Besides analyzing the factors controlling observed WTP, implication of the result should be investigated. Does the rate of 32% WTP is a positive or negative sign to set strategies for better air quality in the site? What can the local government can do with the study result?

Also, a gap between reality and the respondents' perception should be analyzed. The study is totally based on the participants' responds. But it is important whether the respondents' understanding about the air quality is correct. By providing more references of air pollution in the area the fidelity of the interview results can be examined.  For example, what is the relationships between natural gas as the major source of electricity generation in Thailand and air quality of the Mae Moh? In the manuscript, the site is described as the famous for coal mining and power plant. But is that directly related with electricity generation? This kind of reasonings are lacking in the study. 

Finally, sections 1 to 3 (introduction, review of related literature, and materials and methods) need to be rearranged. Especially, some part of section 2 is redundant and can be combined into section 1 and 3. 

Author Response

We thank you immensely for the very useful comments.
Accordingly, we have significantly improved the manuscript.
Below we provide the rejoinder for your kind attention.

Comment 2-1

This study assessed local demand for better air quality in the Mae Moh, Thailand by conducting face-to-face interview and statistical analysis on it.

While the interview result can give valuable results, a clear objective is lacking in logical reasoning. That is, the study found WTP from 32% of the respondents and analyzed factors that can characterize respondents showing WTP but did not gives its meaning that can be applied to policy making.

I would suggest clarify an objective and reorganize the manuscript. Since the interview results are valuable, it can turn into an important paper.

Response 2-1

We appreciate your timely review and providing fair and useful comments. Our responses to your specific comments are found below.

Comment 2-2

I suggest to clarify a goal of the study. Besides analyzing the factors controlling observed WTP, implication of the result should be investigated. Does the rate of 32% WTP is a positive or negative sign to set strategies for better air quality in the site? What can the local government can do with the study result?

Response 2-2

Thank you for the valid comment. Admitting that we had failed to articulate the key result, we have improved the text and clarified that the most important descriptive result is the amounts of WTP, which indicate that the estimated value of clean air was relatively low compared to other studies from major cities being notorious for air pollution. On this basis, we have followed your comment and emphasized the implications of this key result.

First, the low average WTP implies that the FGD installed by the EGAT at the turn of the century had largely eliminated SO2 emissions to date. Second, the low aggregate WTP is partly due to the relatively small population of Mae Moh compared to the metropolitan cities afflicted with air pollution.

On the other hand, while the estimated WTP was relatively low, its non-zero values reflect the remaining and persistent concern over PM pollution as shown by the pollutant specific WTP results, which allegedly poses a risk of developing respiratory symptoms in the long run, if not in the short run. Although mitigation measures are desired, allocation of massive budget cannot be justified from our result unless further evidence arises in support for it.

The revised text can be found in the 1st and 2nd paragraphs of Section 5 and the 3rd paragraph of Section 6.

Comment 2-3

A gap between reality and the respondents' perception should be analyzed. The study is totally based on the participants' responds. But it is important whether the respondents' understanding about the air quality is correct. By providing more references of air pollution in the area, the fidelity of the interview results can be examined. For example, what is the relationships between natural gas as the major source of electricity generation in Thailand and air quality of the Mae Moh? In the manuscript, the site is described as the famous for coal mining and power plant. But is that directly related with electricity generation? This kind of reasonings are lacking in the study.

Response 2-3

Thanks for your valuable comment. Accordingly, we have added additional details in Sections 2.1 and 3.1. In Section 2.1, we have included information regarding the location of the largest (~900 km) natural gas power plant in Thailand, which is also the second largest power plant after Mae Moh. We have also mentioned the natural gas power plant nearest (~600 km) to Mae Moh. Then the geographical reach of major pollutants was discussed. In Section 3.1, we have mentioned the current levels of air pollution in Mae Moh using secondary sources. In Section 5, we have discussed the validity of the estimated WTP with reference to secondary sources.

The revised text can be found in:

the 3rd paragraph of Section 2.1;

the 3rd and 4th paragraphs of Section 3.1; and

the 1st and 2nd paragraphs of Section 5.

Comment 2-4

Finally, sections 1 to 3 (introduction, review of related literature, and materials and methods) need to be rearranged. Especially, some part of section 2 is redundant and can be combined into section 1 and 3.

Response 2-4

Thank you for the valid comment. Accordingly, we have made the following structural revisions:

the 1st paragraph of Section 2.1 has been merged into the 2nd paragraph of Section 1;

the last paragraphs of Section 2.2 and 2.3 have been merged, edited, and became the 1st paragraph of Section 3.3; and

redundancy in Sections 1 and 3 has been removed.

Many thanks and best wishes,